# Mixed-Unit-Model-Based and Quantitative Studies on Groundwater Recharging and Discharging between Aquifers of Aksu River

**Jiyu Huang, Yanyan Ge * and Sheng Li**

The Key Laboratory of Geodynamic Processes and Metallogenic Prognosis of the Central Asian Orogenic Belt College of Geology and Exploration Engineering, Xinjiang University, Urumchi 830017, China; hjyhyj@stu.xju.edu.cn (J.H.); lisheng2997@163.com (S.L.)
* Correspondence: geyanyan0511@xju.edu.cn

**Abstract:** The confined aquifer in the Aksu River Basin is the main aquifer for drinking water within the area. In this study, the unconfined aquifer and the confined aquifer in the Aksu River Basin were divided into different water circulation units through analysis of their flow field. After the hydrochemistry and isotope characteristics of each unit were analyzed, these data were used as water volume quantitative information of the aquifer according to the mixed-unit model. With this quantitative information, the transformation relationship between the unconfined aquifer and the confined aquifer, the recharging source, recharging amount, recharging proportion, and discharging amount of the confined aquifer were revealed. The results showed that the confined aquifer receives a recharge of $21.48 \times 10^6$ m$^3$/a from the unconfined aquifer. The recharging sources of the confined aquifer in the middle and upper stream of the Aksu River mainly included side recharging and leakage recharging from the unconfined aquifer, while the confined aquifer received little recharging from unconfined aquifer downstream of the Aksu River and did not receive recharging from the unconfined aquifer in the southeast of the basin. Additionally, drainage methods of the confined aquifer were mainly lateral flowing and artificial well-group pumping. The side discharging volume through the whole area was $15.67 \times 10^6$ m$^3$/a, and the artificial pumping volume was $21.20 \times 10^6$ m$^3$/a. The confined aquifer was in a negative balance state from the middle-upper stream to the downstream. The downstream confined aquifer and its unconfined aquifer had a plane laminar flow movement, and the unconfined aquifer provided very little recharging to the confined one, which was further enhanced by the artificial well pumping and caused an accumulating negative balance state of the downstream aquifer.

**Keywords:** confined aquifer; unconfined aquifer; transformation; mixed-unit method; Aksu River Basin

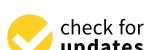

## 1. Introduction

### 1.1. Research Status

As the source of life, water is inseparable from human survival and development. Although there are many water resources, groundwater with stable volume and excellent quality has become an important water resource for agriculture, animal husbandry, industry, and cities. As an indispensable resource for human society, in arid and semi-arid areas with limited precipitation and a small, unevenly distributed surface water resource, the effect of recharging and drainage of groundwater is significant [1,2] and has attracted the attention of many scientists in different fields.

Therefore, in the early 1950s, the United Nations Educational, Scientific and Cultural Organization (UNESCO) began to study the groundwater cycle [3], and the International Association of Hydrogeologists (IAH) also actively carried out many large-scale academic projects on groundwater circulation [4,5]. In China, water resource is the largest and most rigid constraint for production and life throughout the arid area in the northwest. Today,

when we vigorously promote the construction of ecological civilization, we need to insist on using water to plan cities, land, populations, and production. Therefore, the efficient and reasonable development, and sustainable utilization, of groundwater resources are particularly important.

Many methods, such as hydrodynamic methods [6,7], hydrochemical methods [8–10], and environmental isotope methods [11,12] have been reported for studying the relationship between groundwater recharging and discharging.

The hydrodynamic method is based on groundwater chemical dynamics theories, according to the calculation and analysis of hydrochemical indicators (component activity and the mineral saturation index) and limited pumping-test data, five hydrogeological parameters (the permeability coefficient, K; the water conductivity coefficient, T; the actual velocity of groundwater, U; its penetration velocity, v; and the groundwater age, t) and can be used to quantitatively analyze and study all hydrogeological conditions [13,14]. The chemical composition of groundwater is usually controlled by many factors, such as the composition of precipitation, the geological structure, the mineral composition, and the hydrogeological processes of the aquifer. The continuous interaction between groundwater and its surrounding media also changes its chemical composition. Therefore, according to the relative concentration of the main ions in precipitation, surface water, and groundwater from different aquifers, information on the geochemical process in aquifers can be obtained to analyze the law and control mechanism of groundwater evolution, as well as the possible groundwater-evolution path from the recharging area to the discharging area [15–17]. Recently, isotope technology, as a new type of technology, has been developed in hydrogeology to effectively trace the change in water bodies and environment very sensitively, and thus, to record historical information about the evolution of the water cycle [18–20]. Since the 1950s, synthetic isotopes and environmental isotopes have been used to study issues related to hydrology and hydrogeology [21–24]. Many scientists worldwide have used these isotope methods to solve problems related to groundwater recharging resources, surface water transformation, surface-water runoff rate, and the age of surface water. Some scientists have further applied water-chemistry information to groundwater numerical models, used isotopes to trace and determine the recharging resource of groundwater, and calculated the amount of groundwater recharging [25–27]. In the 1990s, quantitative mathematical models became very mature. As one of these mature mathematical models, the mixed-unit model, with water-chemistry data and isotope data, can be used to quantitatively calculate the recharging rate and recharging amount of an aquifer in a specific space [28–31]. These results will be used in studies on groundwater cycles to provide a reliable basis for the rational development and utilization of groundwater resources in the arid area of Northwest China with limited hydrogeological work and low precision.

### 1.2. Purpose of the Research

The Aksu River Basin has four independent rivers from west to east: Aksu River, Kekeya River, Tailan River, and Karayuergun River. The Aksu River is one of the typical large rivers in the northern margin of the Tarim Basin with two major tributaries in its upper stream: the tributary of the Toshigan River on the west and the tributary of the Kumara River on the north. The Toshigan River originates from the Aksai River in the Atbash Mountains of Kyrgyzstan, the Kumarak River comes from the Khan Tengri Peak of the Tianshan Mountains, and both of them recharge rivers with water from snow-melting of glaciers and from precipitation. Twelve km to the south, the Aksu River divides into the Xinda River and the Laoda River. The Laoda River merges into the Xinda River again in Bawutulak, flows south, and enters the Tarim River in Xiaojiake. Its main stream is 132 km long and its drainage area is 63,100 km². The Kekya River originates from the Kochikal Basili Glacier and the Ishtarji Glacier. It goes through the Duolang Canal and merges into the Xinda River in Georgia, and has a total length of 82 km. Both the Tailan River and the Karayuergun River originate from the southern foot of Tuomuer Peak in the South Tianshan Mountains and are independent water systems.

As the Aksu River Basin is located at the southern foot of the Tianshan Mountains and has a dry climate with limited rainfall, the population and agricultural production are currently mainly concentrated in its oasis zone with the confined aquifer as an important water source. Many scientists have studied and provided information on the transformation relationship between river and groundwater (mostly unconfined aquifers). However, the recharging and discharging relationships, and the circulation mode of the confined aquifer are not currently understood.

In this study, based on data from the unconfined aquifer and confined aquifer flow fields in the basin, samples of river water, the unconfined aquifer, and the confined aquifer were systematically collected. After their water chemistry and isotope distribution characteristics were analyzed, the mixed-unit method was used to quantify these data, and thus, to reveal the recharging source and circulation mode of the confined aquifer.

## 2. Geology and Hydrogeology

The Aksu River Basin is located at the southern foot of Tianshan Mountains and the northern edge of the Tarim Basin, which belongs to the first-level tectonic unit of the Tarim platform. The water system in this basin was formed from the end of the Tertiary to the beginning of the Quaternary. Due to the neotectonic movement of the northern mountain body, a downstream river system was formed along the south-dipping slope of the mountain body. The water flow has brought mountain debris to the front of the mountain and deposited it in the Awati fault depression, gradually forming the alluvial plain of the Aksu River and the Kekeya River. Additionally, the uplift of the Yingan Mountains has led to a decline of the southeast side of the study area, and the formation of a strip of lowland in Aiximan (Figure 1). With water accumulation, a bead-like lake group was generated. Meanwhile, the Aksu River continued to swing in periods and moved eastward to the current Laoda River and Xinda River, leaving several river traces in the west of the plain, which then evolved into an intermittent strip of an oxbow lake, as shown in Figure 1. The geomorphological units of the Aksu River Basin from north to south are the piedmont alluvial fan group, the alluvial–proluvial slope plain, and the fine-soil-grain plain. As shown in Figure 2, from north to south, the lithology changes from coarse to fine, and sandy gravel changes from medium-coarse sand, to fine sand, to sandy loam. The sloping gravel plain area in the piedmont zone is a single unconfined aquifer area. Its water is more than 50 m in depth with the deepest part being 220 m, and its thickness is 90–100 m. The gently sloping fine-soil plain area and the desert plain area are a multi-layer area with unconfined and confined aquifers. The unconfined aquifer of Tumuxiuke Town–Wensu-Jiamu Town, north of Wutuan is buried 10–50 m deep, and the middle and downstream of the unconfined aquifer of the alluvial plain is less than 10 m in depth. In the south of Ayikule Town, Rice Farm, and the south of Wutuan, groundwater overflows from an artesian well. The south and southeast are formed with confined aquifer rock groups (mainly sand layers), and the thickness of the confined aquifer gradually increases from the north to the south within 15–130 m. The confined aquifer winging out in the west of Aksu is influenced by the Yinganshan uplift. The groundwater flow in the unconfined aquifer and the confined aquifer in the Aksu River Basin is affected by this neotectonic movement, and flows from north to south. Its downstream flowing direction changes from north-to-south to south-to-east as shown in Figures 3 and 4.

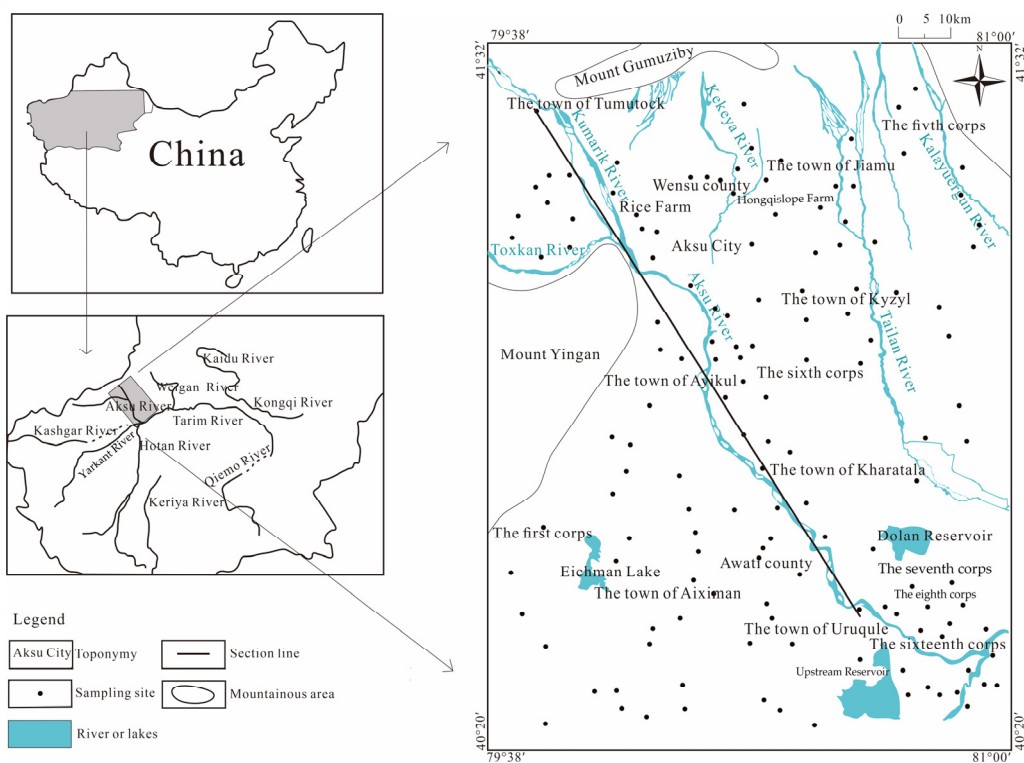

**Figure 1.** Locations of the studied area and sampling sites.

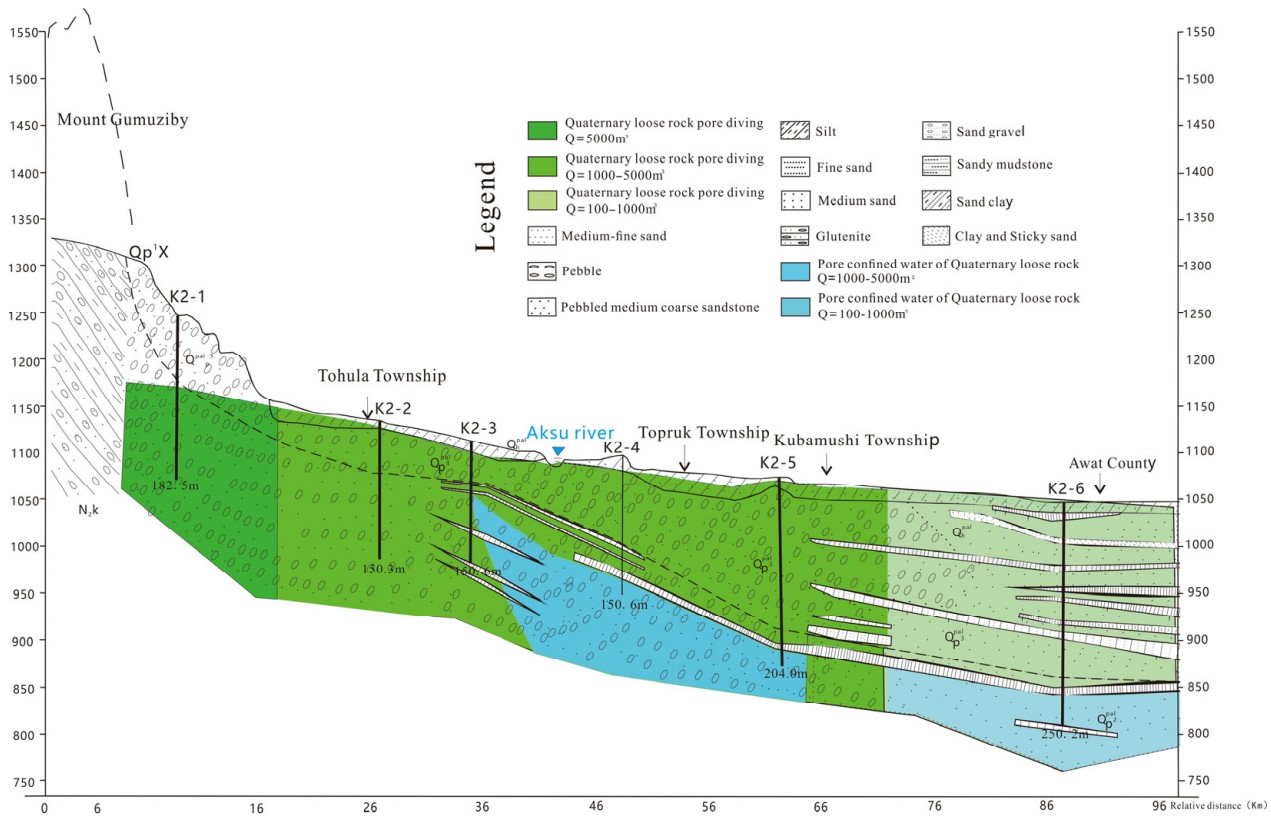

**Figure 2.** The hydrogeological profile of A–A′ section.

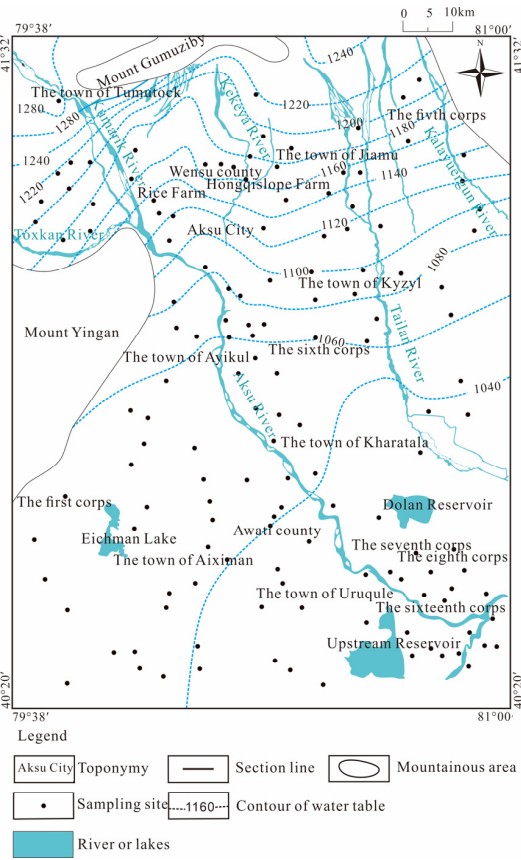

**Figure 3.** Contour lines of unconfined aquifer.

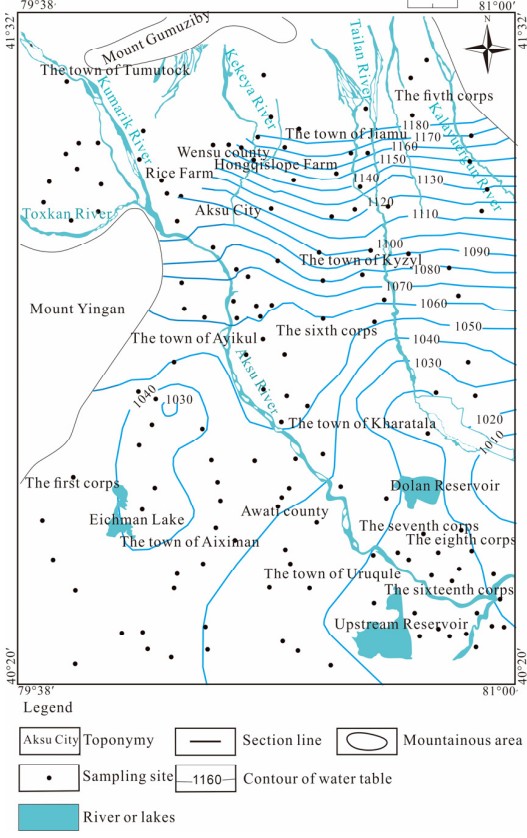

**Figure 4.** Contour lines of confined aquifer.

### 3. Sample Collection and Testing

A total of 196 groups of water samples were collected, including 151 groups from the unconfined aquifer and 45 groups from the confined aquifer. There are 23 groups of environmental isotope samples, including 15 groups from the unconfined aquifer and 9 groups from the confined aquifer. Sampling locations are shown in Figure 5.

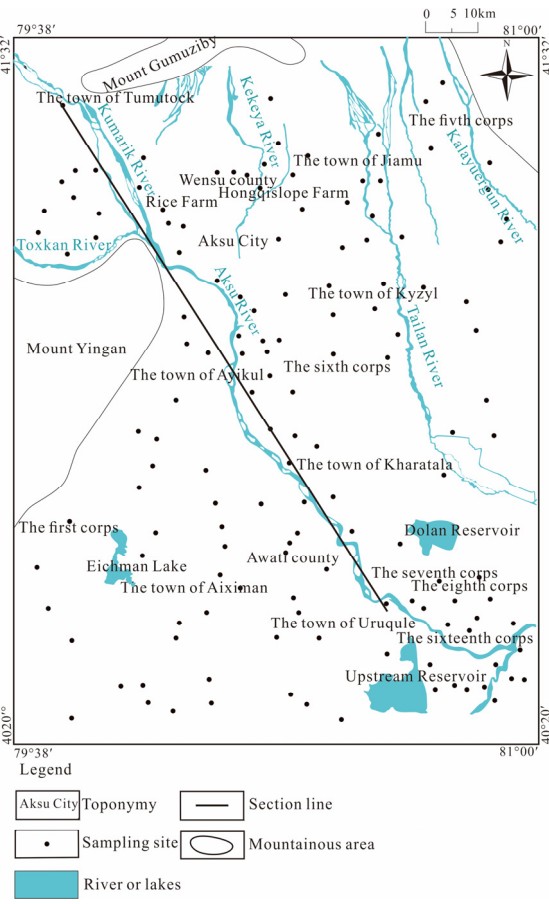

**Figure 5.** Sampling locations in the area.

The collected samples were analyzed by the first regional geological survey team of the Xinjiang Geological and Mineral Bureau to determine $K^+$, $Na^+$, $Ca^{2+}$, $Mg^{2+}$, $HCO_3^-$, $CO_3^{2-}$, $SO_4^{2-}$, and $Cl^-$ on an inductively coupled plasma spectrometer and an atomic absorption spectrophotometer with accuracy of $\pm 0.5\%$ and $\pm 1\%$. Environmental isotope samples were analyzed in the American BETA laboratory to determine $\delta D$ and $\delta^{18}O$ with accuracy of $\pm 2\%$ and $\pm 0.3\%$, respectively, on an isotope mass spectrometer (Thermo Delta-Plus) after high-temperature treatment, evaporation, dissociation, atomization, and ionization.

### 4. Principles and Theory of Mixed-Unit Method

#### 4.1. Hypothesis of Mixed-Unit Method

In mixed-unit method, the aquifer is generalized and discretized into a finite number of homogeneous and isotropic small units. Each small unit has a comprehensive value to show its hydrochemical characteristics (ion concentration and isotope value). According to their flow fields, the possible recharging and discharging relationship is obtained. With the ion concentration and isotope value in each unit as its tracer, the tracer mass-conservation equation can be established. Through solving this equation, the recharging and discharging relationships and recharging ratio can also be determined. Before the determining of the mixed units, the following assumptions need to be made: (1) in order to qualitatively judge the groundwater charging and discharging conditions, the tracer concentration of the water

resource and the discharged water flow are already known; (2) conservation of water level: in each small unit, within a certain time, the water level is constant, and the water level is averaged; (3) the migration of dissolved components is controlled by convection; and (4) effects of mineral reaction, dissolution, and precipitation are negligible.

### 4.2. Unit Determination Principles

In order to reduce the unknown parameters of the model and determine the small units in an optimal way, the following principles need to be followed: (1) The studied area is divided along the groundwater flow into units, whose horizontal unit boundary must be parallel to the groundwater level contour line and longitudinal boundary must be perpendicular, or approximately perpendicular, to the groundwater level contour line; (2) A hydrogeological unit can be divided into multiple small units. A small unit cannot cross into different hydrogeological units. Different hydrogeological units store different types of groundwater with different ion composition and isotope values. (3) A small unit should have representative water sample data. (4) The same cone of depression should be divided into one small unit.

### 4.3. Unit Determination

According to the above-mentioned assumptions and principles of the mixed-unit method, the unconfined aquifer of the Aksu River Basin was divided into seven small units (a, b, c, d, e, f, and g), and its confined aquifer is divided into five small units (C, D, E, F, and G), as shown in Figure 6.

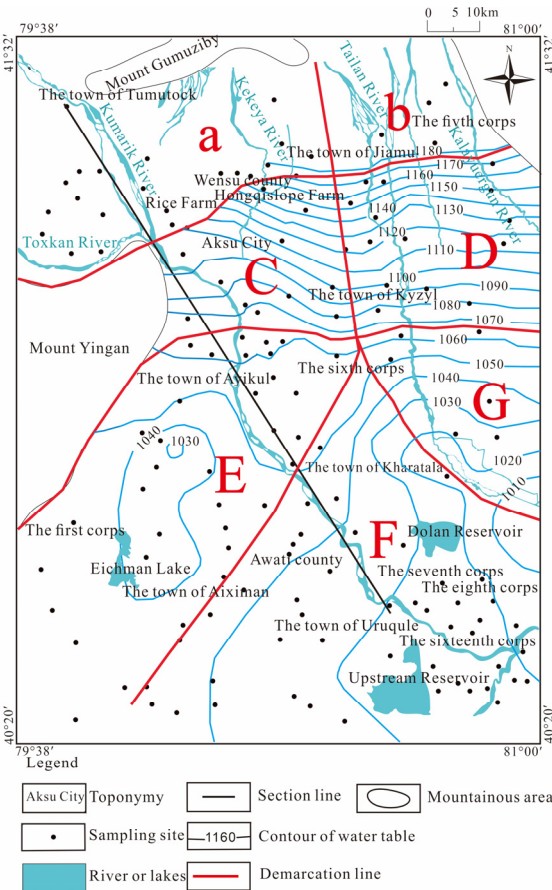

**Figure 6.** Mixed-unit division in the confined aquifer.

*4.4. Calculation of Aquifer Recharging and Discharging*

In the mixed-unit model, the unconfined aquifer and confined aquifer are divided into a finite number of discrete small units, which are discrete at an interval of Δt. Their solutes are fully mixed, and the components of each solute are evenly distributed in all small units. Therefore, the equilibrium equation of water in a small unit within the period of Δt can be expressed as:

$$Q_n - W_n + \sum_{i=1}^{In} q_{in} - \sum_{i=1}^{In} q_{nj} = e_n \tag{1}$$

According to the assumption on the water balance of each small unit, the mass balance equation of the dissolved component k in unit n can be obtained as:

$$C_{nk}Q_n - C_{nk}\left[W_n + \sum_{j=1}^{In} q_{nj}\right] + \sum_{i=1}^{In} q_{in}C_{ink} = e_{nk} \quad k = 1, 2, \ldots, k \tag{2}$$

where, $Q_n$ represents the time average flow value into unit n, $W_n$ is the average value of the flow out from unit n, $q_{in}$ represents the average flow from unit i into n, $e_n$ is the deviation of water balance caused by various errors from the flow entering or exiting the unit n, k is the average concentration of the tracer k in one unit, and $C_{nk}$ is the average concentration of the trace k in the k in unit n.

After Equations (1) and (2) are combined into a rectangular matrix of known concentrations in unit n, in which the first row represents the water balance and the other rows represent the solute mass conservation balance, the Equation (3) can be obtained with any unit n:

$$C_n q_n + D_n = E_n \tag{3}$$

where, $q_n$ represents the flow through the boundary of small unit n:

$$q_n = [q_{1n}q_{2n}\cdots q_{in}q_{n1}q_{n2}\cdots q_n j_n](In + Jn) \times 1 \tag{4}$$

$D_n$ is the measurable and quantifiable known items in unit n (such as the known outflow and pumping volumes), and $E_n$ represents the unknown error vector in the unit as,

$$E_n = [e_n e_{n1} e_{n1} \ldots e_{nk}](1 + K) \times 1 \tag{5}$$

According to Equation (3) (Adar (1988)), through the minimization of the sum function J of square error and evaluation of the sum of square error of all units, the flow composition of the aquifer can be obtained as,

$$J = \sum_{1}^{N}[E_n^T W E_n] = \sum_{1}^{N}(c_n q_n + D_n)^T W(c_n q_n + D_n) \tag{6}$$

## 5. Calculation of Charging and Discharging of Confined Aquifer with Mixed-Unit Method

*5.1. Division of Mixed Units*

Because the studied area is located in the plain of alluvial–diluvial fine-soil particles, the conceptual model of the mixed units was established accordingly as shown in Figure 5. The mixed units of the unconfined aquifer and confined aquifer were, respectively, marked as a, b, c, d, e, f, and g and C, D, E, F, and G. Units a and b are located in the alluvial–proluvial slope gravel plain as a single-structure unconfined aquifer and are the recharging source of confined aquifer. All other units are in the alluvial–diluvial fine-soil-grain plain. Units G and F are discharging units. The transformation relationship between units of the aquifer is shown in Figure 7.

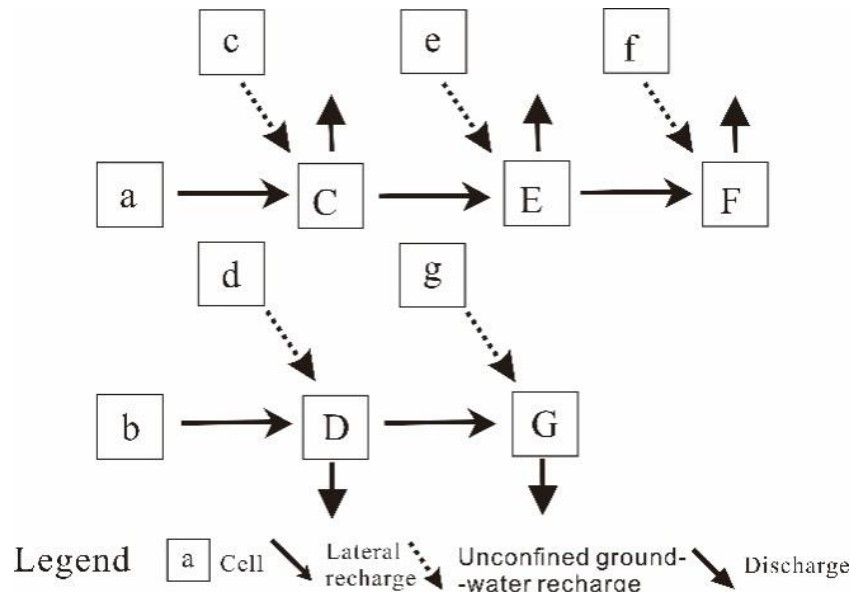

**Figure 7.** The transformation relationships between units.

*5.2. Hydrochemical Characteristics*

The unconfined aquifer is widely distributed in the studied area, and its salinity in the upper stream was about 0.6 g/L, and pH was 8.4 with an $HCO_3 \cdot SO_4$–$Mg \cdot Ca$-type water. It was 1.3–3 g/L in the middle stream with water types of $SO_4 \cdot Cl$–$Na \cdot Mg$ and $SO_4 \cdot Cl$–$Na \cdot Ca \cdot Mg$, and pH was 8.3. The salinity in the west of the downstream study area was 3.7 g/L, pH was 8.4, and the salinity in the east was 9.0 g/L, pH was 8.2, with water types of $Cl \cdot SO_4$–$Na$ and $Cl$–$Na$. The water-chemistry type of unconfined aquifer units a→c→e→f changed from $HCO_3 \cdot SO_4$–$Mg \cdot Ca$, to $SO_4 \cdot Cl$–$Na \cdot Mg$, to $SO_4 \cdot Cl$–$Na \cdot Ca \cdot Mg$, to $Cl \cdot SO_4$–$Na$ (or $Cl$–$Na$. That of unconfined aquifer b→d→g changed from $HCO_3 \cdot SO_4$–$Mg \cdot Ca$ to $Cl \cdot SO_4$–$Na$ (or $Cl$–$Na$) and $SO_4 \cdot Cl$–$Na \cdot Mg$ (or $SO_4 \cdot Cl$–$Na \cdot Ca \cdot Mg$).

The salinity of confined aquifer from the middle-upper stream to the downstream of the Aksu River Basin did not change significantly with a salinity of 1 g/L, pH of 8.1 and water types of $SO_4 \cdot Cl$–$Ca \cdot Na \cdot Mg$, $SO_4 \cdot HCO_3 \cdot Cl$–$Mg \cdot Na \cdot Ca$ and $Cl \cdot SO_4$–$Na \cdot Ca$ in the downstream. Water types of the confined aquifer unit C→E→F changed from $SO_4 \cdot Cl$–$Ca \cdot Na \cdot Mg$ or $SO_4 \cdot HCO_3 \cdot Cl$–$Mg \cdot Na \cdot Ca$ to $Cl \cdot SO_4$–$Na \cdot Ca$. From units D→G, it changed from $SO_4 \cdot Cl$–$Ca \cdot Na \cdot Mg$ or $SO_4 \cdot HCO_3 \cdot Cl$–$Mg \cdot Na \cdot Ca$ to $Cl \cdot SO_4$–$Na \cdot Ca$. These data showed that, along the flowing path of the confined aquifer, in the middle and upper steams of the west, the confined aquifer receives a large amount of recharging laterally from the unconfined aquifer (a→C→E), and a small amount of recharging vertically from the unconfined aquifer (c→C→E). When the water exchange between the confined aquifer and the unconfined aquifer was reduced (e→E and f→F), evaporating concentration and cation-exchange adsorption (e→f) occurred in the unconfined aquifer, and cation exchange adsorption occurred in the confined aquifer (E→F). The middle and upper streams in the east receive a large amount of recharging water vertically from unconfined aquifer (d→D), with a small amount from unconfined aquifer laterally (b→D). Its downstream receive a large amount of water recharging laterally from confined aquifer (D→G). with a small amount vertically from unconfined aquifer (g→G). The unconfined aquifer has significant evaporating concentration (d→g), and the confined aquifer mostly has cation exchange adsorption (D→G). The water chemistry characteristics in this basin are shown in Figure 8.

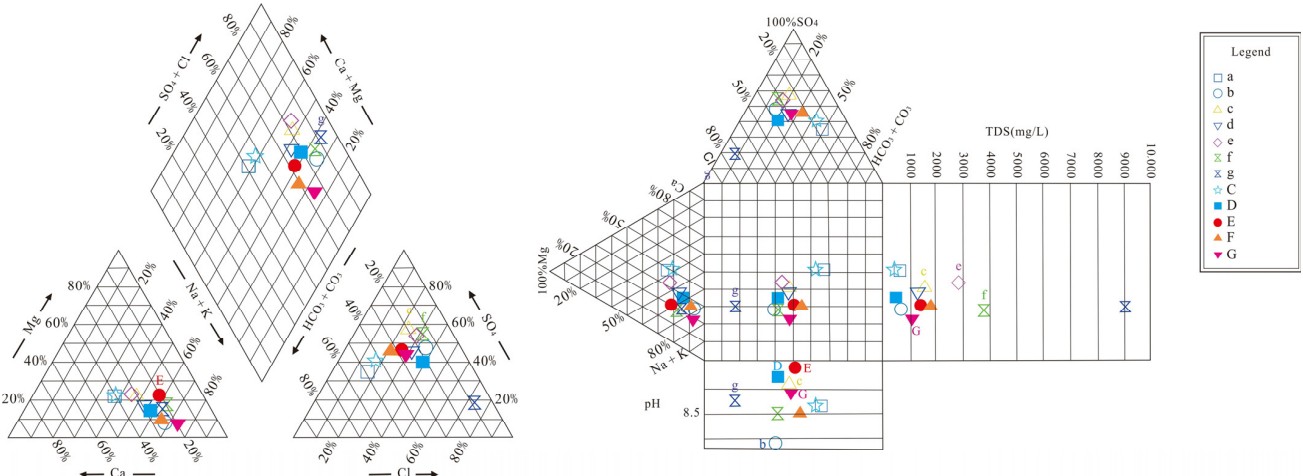

**Figure 8.** Water chemistry characteristics.

## 5.3. Distribution Characteristics of Isotopes

As shown in Figure 9, after the precipitation line slope of the aquifer of the studied area was compared with the Global Meteoric Water Line (GMWL) proposed by Craig [32], the slope and intercept were both smaller than the global precipitation line and $\delta D$ and $\delta^{18}O$ were all located at the upper left of GMWL. These results showed that strong evaporation had taken place in the whole of the aquifer.

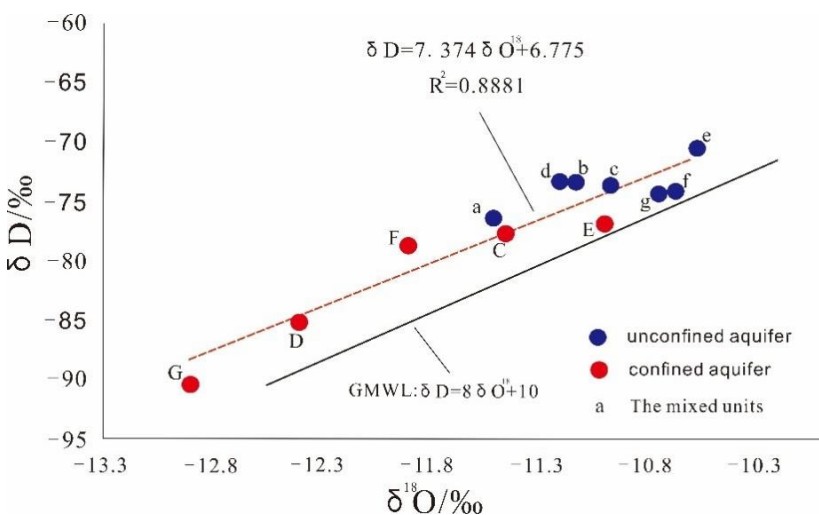

**Figure 9.** Relationship between $\delta D$ and $\delta^{18}O$.

The isotopic compositions of the unconfined aquifer(a→c→e→f) and the confined aquifer(C→E→F) in the west were basically the same, indicating that unconfined aquifer and confined aquifer had a certain hydraulic connection. As shown in Figure 10a, in the middle and upper steams of the west, because of the influence of evaporation, the $\delta D$ and $\delta^{18}O$ in the unconfined aquifer increased gradually, while the $\delta D$ and $\delta^{18}O$ in the confined aquifer increased simultaneously. These results showed that the unconfined aquifer recharged the confined aquifer vertically (c→C, e→E). In the downstream, the $\delta D$ and $\delta 18O$ of the unconfined aquifer f and the confined aquifer F were significantly different, indicating that the confined aquifer was less replenished by the unconfined water ( f→F). The $\delta D$ and $\delta^{18}O$ of unit C→E→F were closer to unit c→e→f than to unit a, indicating that the confined aquifer received a large amount water recharging laterally from the unconfined aquifer (a→C→E→F).

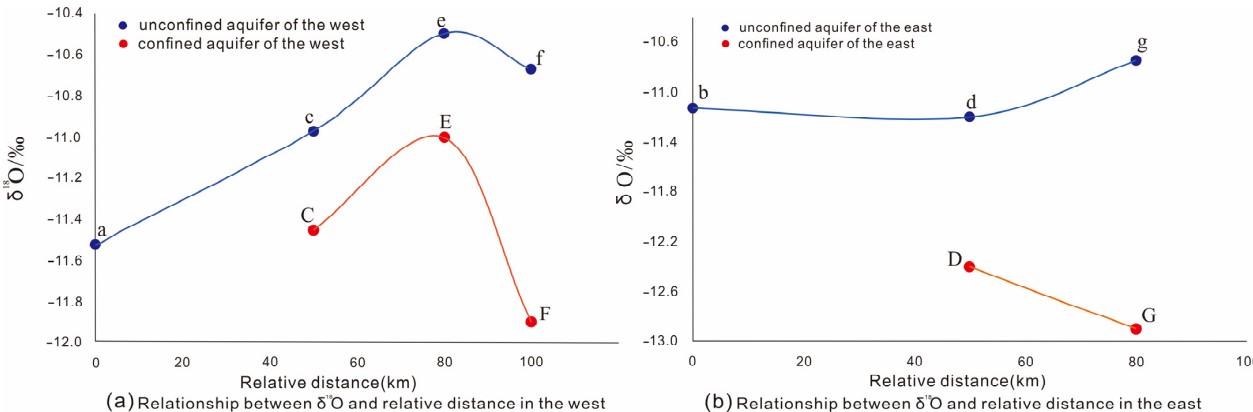

**Figure 10.** Relationship between $\delta^{18}$O and relative distance.

The δD and $\delta^{18}$O values of the unconfined aquifer (b→d→g) and the confined aquifer (D→G) in the east were significantly different, indicating that the hydraulic connection between the two water bodies was weak. As illustrated in Figure 10b, from upper to lower in the east, because of the influence of evaporation, the δD and $\delta^{18}$O were enriched along the way and in the confined aquifer, the δD and $\delta^{18}$O decreased. These results showed that the confined aquifer in the middle and upper streams received a large amount of water recharging vertically from the unconfined aquifer (d→D) and the confined aquifer in the lower streams received a small amount of water recharging vertically from unconfined aquifer (g→G).The δD and $\delta^{18}$O of unit D→G were closer to Unit d→g than to the mountain exit stations, indicating that the confined aquifer D→G received a large amount water recharging laterally from unconfined aquifer (b→D→G).

### 5.4. Recharging Relationships between Unconfined Aquifer and Confined Aquifer

According to the hydrochemical and isotope distribution characteristics of the studied area, the confined aquifer in the alluvial–diluvial plain area received water recharging from the loose rock porous unconfined aquifer and upper porous unconfined aquifer of the Quaternary in the upper alluvial–diluvial slope plain, which was discharged through artificial and lateral downstream. The recharging and discharging relationship between units in the unconfined aquifer and confined aquifer are shown in Figure 11.

**Figure 11.** Groundwater circulation in the unconfined aquifer and the confined aquifer.

### 5.5. Data Analysis

The hydrogeochemical and isotopic data of each unit were statistically analyzed and are listed in Table 1.

**Table 1.** Hydrochemical properties and isotopes of all units.

| Unit | Item | $K^+$ | $Na^+$ | $Ca^{2+}$ | $Mg^{2+}$ | $Cl^-$ | $SO_4{}^{2-}$ | $HCO_3{}^-$ | D | $^{18}O$ |
|---|---|---|---|---|---|---|---|---|---|---|
| a | Average | 0.14 | 2.39 | 3.25 | 2.95 | 1.95 | 2.98 | 3.21 | −76.36 | −11.52 |
| | Mean square error | 3.65 | 7.70 | 4.23 | 3.53 | 5.93 | 1.20 | 25.52 | 6.65 | 0.44 |
| b | Average | 0.18 | 9.76 | 4.25 | 2.41 | 7.27 | 6.99 | 1.58 | −73.44 | −11.13 |
| | Mean square error | 0.93 | 13.59 | 3.60 | 1.02 | 0.77 | 13.72 | 4.99 | - | - |
| c | Average | 0.21 | 8.98 | 6.76 | 8.06 | 7.41 | 12.89 | 3.40 | −73.72 | −10.97 |
| | Mean square error | 0.33 | 5.28 | 2.40 | 3.73 | 5.93 | 0.01 | 3.85 | 0.96 | 0.03 |
| d | Average | 0.12 | 5.18 | 3.45 | 3.40 | 4.46 | 4.96 | 2.03 | −73.34 | −11.20 |
| | Mean square error | 2.79 | 3.82 | 7.18 | 2.95 | 2.18 | 8.02 | 8.24 | - | - |
| e | Average | 0.19 | 7.56 | 6.50 | 7.56 | 8.24 | 10.87 | 2.66 | −70.57 | −10.50 |
| | Mean square error | 2.35 | 11.14 | 4.76 | 6.00 | 11.84 | 1.72 | 25.52 | 7.67 | 0.49 |
| f | Average | 0.23 | 19.46 | 6.82 | 10.55 | 14.61 | 18.46 | 3.50 | −74.2 | −10.67 |
| | Mean square error | 11.6 | 0.09 | 9.14 | 0.95 | 1.98 | 11.17 | 0.01 | 5.29 | 0.11 |
| g | Average | 0.12 | 11.36 | 4.69 | 5.20 | 15.82 | 2.92 | 0.98 | −74.38 | −10.75 |
| | Mean square error | 0.06 | 11.17 | 0.38 | 0.16 | 0.16 | 5.90 | 20.86 | - | - |
| C | Average | 0.15 | 2.56 | 3.43 | 3.15 | 2.21 | 3.76 | 3.07 | −77.70 | −11.45 |
| | Mean square error | 0.75 | 9.88 | 1.27 | 1.58 | 3.85 | 0.85 | 1.88 | 9.61 | 0.20 |
| D | Average | 0.09 | 4.64 | 2.66 | 2.30 | 4.38 | 3.30 | 1.40 | −85.20 | −12.40 |
| | Mean square error | 1.84 | 11.73 | 0.15 | 6.51 | 8.63 | 6.39 | 24.35 | - | - |
| E | Average | 0.10 | 4.70 | 1.96 | 3.32 | 3.40 | 4.39 | 2.17 | −76.90 | −11.00 |
| | Mean square error | 4.62 | 7.76 | 0.31 | 15.54 | 5.93 | 1.60 | 25.52 | - | - |
| F | Average | 0.06 | 3.32 | 1.56 | 0.90 | 1.83 | 2.50 | 1.30 | −78.70 | −11.90 |
| | Mean square error | 0.94 | 3.26 | 0.57 | 4.58 | 29.60 | 8.14 | 3.85 | - | - |
| G | Average | 0.06 | 4.92 | 1.51 | 0.93 | 2.65 | 3.05 | 1.38 | −90.40 | −12.90 |
| | Mean square error | 2.79 | 9.52 | 5.11 | 0.59 | 11.85 | 10.40 | 22.69 | - | - |

Notes: (1) Concentrations are in meq/L unless otherwise indicated, such as deuterium and oxygen $^{18}O$ in ‰, and (2) "-" means that there is only one sampling point and the square error cannot be determined.

In the studied area, the drainage methods of the confined aquifer were mainly lateral runoff discharging and artificial well-group pumping. Artificial exploitation was $27.95 \times 10^6$ m$^3$/a and the mining volume of each unit was calculated according to the proportion of the unit area. The mining volume of each unit is shown in Table 2. The lateral discharging volumes were $8.57 \times 10^6$ m$^3$/a and $7.10 \times 10^6$ m$^3$/a, as shown in Table 3.

**Table 2.** Groundwater mining volume of each unit.

| Unit | a | b | C | D | E | F | G |
|---|---|---|---|---|---|---|---|
| Surface area (km$^2$) | 724.72 | 400.05 | 988.55 | 655.2 | 810.18 | 714.27 | 365.34 |
| Mining volume ($10^6$ m$^3$/a) | 4.35 | 2.4 | 5.93 | 3.93 | 4.86 | 4.29 | 2.19 |

**Table 3.** Groundwater discharging laterally from Units F and G.

| Unit | Buried Depth of Groundwater (m) | Aquifer Thickness (m) | Osmotic Coefficient (m/d) | Hydraulic Gradient | Discharging Volume ($10^6$ m$^3$/a) |
|---|---|---|---|---|---|
| F | 4.03 | 180 | 2.5 | 1/1050 | 8.57 |
| G | 1.88 | 180 | 2 | 1/740 | 7.1 |

### 5.6. Results and Analysis

The results showed (Figure 12) that units a and b in the studied area were single-structure unconfined aquifers as the main recharge resources of the downstream confined aquifer. Unit C accepted the lateral recharge from unit a and the leakage recharge from Unit c, which were $9.77 \times 10^6$ m$^3$/a and $0.64 \times 10^6$ m$^3$/a, respectively. Unit E received the lateral recharge from unit C and the overflow recharge from unit e, which were $4.67 \times 10^6$ m$^3$/a and $2.80 \times 10^6$ m$^3$/a, respectively. The unit F received the lateral recharge from the unit E and the overflow recharge from the f unit, which were $6.10 \times 10^6$ m$^3$/a and $0.90 \times 10^6$ m$^3$/a, respectively. Unit D received the lateral recharge from unit b and the overflow recharge from unit d, which were $1.71 \times 10^6$ m$^3$/a and $5.66 \times 10^6$ m$^3$/a, respectively. The unit G received the lateral recharge from the unit D, and the recharging was $5.90 \times 10^6$ m$^3$/a. This recharge relationship and degree were consistent with results from hydrochemistry and isotope.

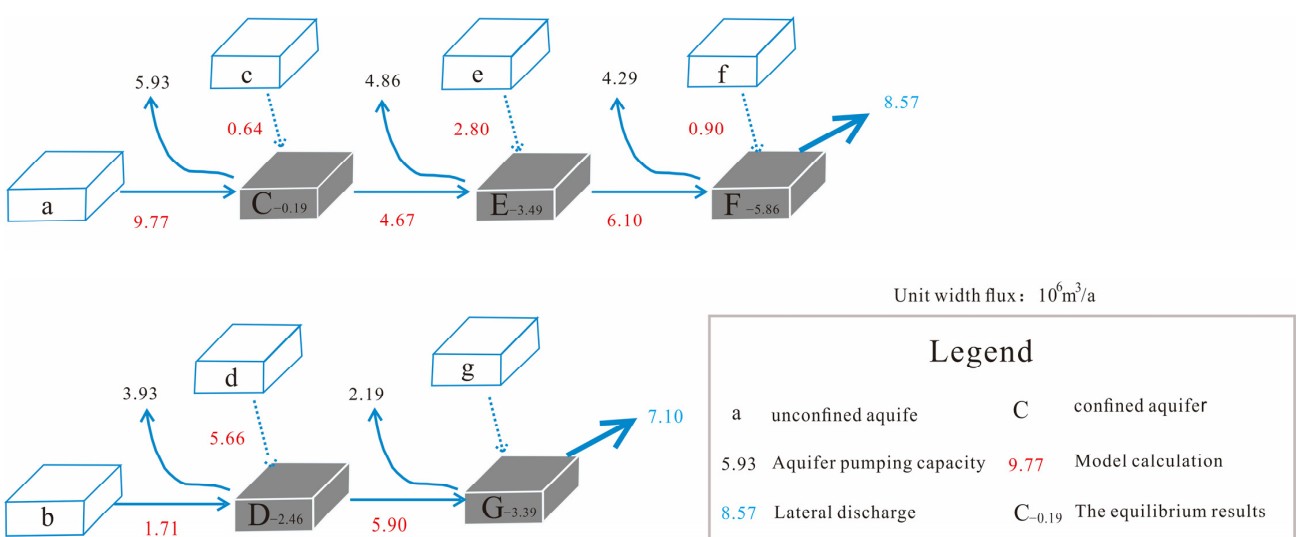

**Figure 12.** Quantitative water circulation model of the unconfined aquifer and the confined aquifer in the studied area.

The total groundwater recharge volume from the confined aquifer in the studied area was $21.48 \times 10^6$ m$^3$/a, of which the lateral recharge was $11.48 \times 10^6$ m$^3$/a, accounting for 53.45% of the total, and the vertical leakage recharge was $10.00 \times 10^6$ m$^3$/a, accounting for 46.55% of the total. The vertical leakage recharge in the southeast was very small and negligible. The total discharging volume from the confined aquifer was $36.87 \times 10^6$ m$^3$/a, of which the lateral discharging volume was $15.67 \times 10^6$ m$^3$/a, accounting for 42.5%, and artificial exploitation was $21.2 \times 10^6$ m$^3$/a, accounting for 57.5%. These results showed that the confined aquifer was in an accumulation and superposition state of negative balance along the direction of the underground water flow.

### 5.7. Discussion

Based on the results of the water balance and the model output, the upstream confined aquifer received lateral recharging and vertical leakage recharging from the unconfined aquifer, and the downstream confined aquifer only received lateral recharging from the upstream confined aquifer, which was consistent with the hydrochemical and isotope analysis. In addition, the unit flux between the cells was apparently influenced by the water source exploitation. For example, the unit E received the recharge from unit e and unit C was $7.47 \times 10^6$ m$^3$/a, the discharged through lateral was $6.10 \times 10^6$ m$^3$/a. Because of over extraction, the difference between the inflow and outflow to unit E was $-3.49 \times 10^6$ m$^3$/a. Compared with previous research, we identified the recharge sources and their relative contributions to the confined aquifer, and the data of the extraction were available, the quantitative water circulation model were reasonable, and the results were reliable.

## 6. Conclusions

According to the geological and hydrogeological data of the studied area, the recharging and discharging relationship between the unconfined aquifer and the confined aquifer was determined. Based on the hypothesis and principles of unit dividing, and the qualitative recharging and discharging relationship, a mixed-unit model was established to study the hydrochemical characteristics of the confined aquifer in the west of the studied area. Results showed that the confined aquifer was significantly affected by unconfined aquifer, and the confined aquifer received lateral recharging from upstream of the unconfined aquifer and vertical recharging from the upper unconfined aquifer. In the east of the studied area, the downstream confined aquifer no longer received the recharging vertically from unconfined aquifer, but mainly received the lateral recharging from the upstream confined aquifer.

The $\delta D$ and $\delta^{18}O$ of unconfined aquifers in the studied area were both at the upper left of the Global Meteoric Water Line, and their slope and intercept were both smaller than those of the global atmospheric precipitation line, indicating that the unconfined aquifer was significantly evaporated. The isotopic compositions of unconfined aquifer and confined aquifer in the west of the studied area were basically the same, indicating that there was a hydraulic connection between these two water bodies, and the confined aquifer received lateral recharging from the upstream unconfined aquifer and vertical leakage recharging from the upper unconfined aquifer. The $\delta D$ and $\delta^{18}O$ distributions of the unconfined aquifer and confined aquifer in the east of the studied area were relatively discrete, indicating that the hydraulic connection between these two water bodies was weak, and the downstream of confined aquifer in the east mainly received lateral recharging from the upstream confined aquifer. These results were consistent with the hydrochemical analysis.

Based on the mixed-unit model, the calculation results showed that the total recharged volume received by the confined aquifer in the studied area was $21.48 \times 10^6$ m$^3$/a, in which the lateral recharging was $11.48 \times 10^6$ m$^3$/a, accounting for 53.45% of the total, and the vertical recharging was $10.00 \times 10^6$ m$^3$/a, accounting for 46.55% of the total. The vertical recharging amount in the southeast was very small and negligible. The total discharging volume was $36.87 \times 10^6$ m$^3$/a, including lateral discharging whose amount was $15.67 \times 10^6$ m$^3$/a, accounting for 42.5%, and the artificial pumping amount was $21.2 \times 10^6$ m$^3$/a, accounting for 57.5%. The upstream confined aquifer received lateral recharging and vertical leakage recharging from the unconfined aquifer, and the downstream confined aquifer only received lateral recharging from the upstream confined aquifer. The confined aquifers in the entire region were in a state of negative balance, and this state was continuously accumulated from the upstream to the downstream.

**Author Contributions:** Conceptualization, J.H. and Y.G.; methodology, J.H.; software, J.H.; validation, J.H., Y.G. and S.L.; formal analysis, J.H.; investigation, J.H.; resources, Y.G.; data curation, Y.G.; writing—original draft preparation, J.H.; writing—review and editing, J.H. and Y.G. All authors have read and agreed to the published version of the manuscript.

**Funding:** This study was supported by grants from the National Fund Program (No. U1603243), and the Key Laboratory of Geodynamic Processes and Metallogenic Prognosis of the Central Asian Orogenic Belt (No. 2020–004).

**Institutional Review Board Statement:** Not applicable.

**Informed Consent Statement:** Not applicable.

**Data Availability Statement:** The data that support the findings of this study are available from the corresponding author upon reasonable request.

**Conflicts of Interest:** The authors declare no conflict of interest.

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
