# Peer review of "Mixed-Unit-Model-Based and Quantitative Studies on Groundwater Recharging and Discharging between Aquifers of Aksu River"

_sustainability, doi:10.3390/su14116936_

Round 1

Reviewer 1 Report

The article bearing the title"Mixed-Unit-Model-Based and Quantitative Studies on Ground Water Recharging and Discharging between Aquifers of Aksu  River" is very interesting for the readers. The article can be accepted after the revision as given below

Figure 1. Locations of the studied area and sampling sites, it is required to modify with an elevation of the study area and stream network

All figures must be high resolution as well as readable all the parameters

Also give the evidence of sample collection and testing in form figure in section 3, which attract the readers more and increase the cition of article

Discuss how much results will deviate if assumptions were not made?

Was the model calibrated and validated? If yes, mention the sensitivity parameters most affected during calibration.

Discuss whether model results were compared with some other model for validation?

The discussion section is missed in the article, please compare the results with other studies.

Reviewer 2 Report

1. Introduction

The work is interesting, and the authors have well described theoretical background and conducted approach.

The paper does not follow the usual scientific paper structure (Introduction - Methods -Results - Analysis…) but in this case I do not mind because it is a demonstration of the use of mixed-unit method, so breaking it into predefined units might make it difficult to read the paper.

As for the content itself, I have no important remarks, but only some of the suggestions that I listed below.

Abstract

Pay attention to the way units and values ​​are displayed (Line 18, Line 24-25)

1. Introduction

In the introduction, a good link is made between previous method related to relationship between groundwater recharging/discharging and what is new in this approach. This is also supported by an adequate number of references.

I suggest that the text be broken into smaller sections, so that it can be more easily followed, especially separating the part describing previous research and settings from the part related to the description of the research area (Line 83 onwards).

2. Geology and hydrogeology

Geological and hydrogeological characteristics are well described. I suggest that the exact data be further supported by the necessary references, eg Line 127-128, line 131-132, line 136.

Figure 1. Indicate on the position sketch that it is China. In the sketch with sampling sites, along the lines / polygons showing watercourses, state the names of watercourses

Figure 2. Improve the visibility of the legend - by increasing the font. I also recommend using a hatch for hydrogeological / geological rock types because in the case of a black and white figure, nothing will be different.

Figure 3, Figure 4. Indicate the contour lines in the Legend. Also consider different lines so that they can differ from each other even in the case of a black and white display. Next to the lines / polygons showing the watercourses, state the names of the watercourses

3. Sample collection and testing

I have no objections.

4. Principles and theory of Mixed Unit Method

Figure 5. In the Legend, indicate the contour lines. Also consider different lines so that they can differ from each other even in the case of a black and white display. Next to the lines / polygons showing the watercourses, state the names of the watercourses.

5. Calculation of charging and discharging of confined aquifer with Mixed Unit Method

Figure 7. Use larger displays for better resolution because markings are poorly visible. If the legend is the same for both diagrams, then use only one and increase it.

Line 269. I don't know where Figure 8-a is. Probably the authors meant Figure 8.

Line 277. Typo: Because> because

6. Conclusions

I have no objections. Conclusions thoroughly supported by the results presented in the article

References

The article is adequately referenced.

English language and style

I don’t feel qualified to judge about the English language and style because I am not a native English speaker, but everything seems understandable to me and I feel that no major interventions in the text are needed.
